# Therapeutic Effect of Biomimetic Scaffold Loaded with Human Amniotic Epithelial Cell-Derived Neural-like Cells for Spinal Cord Injury

**DOI:** 10.3390/bioengineering9100535

**Published:** 2022-10-09

**Authors:** Chen Qiu, Yuan Sun, Jinying Li, Yuchen Xu, Jiayi Zhou, Cong Qiu, Shaomin Zhang, Yong He, Luyang Yu

**Affiliations:** 1MOE Laboratory of Biosystems Homeostasis & Protection of College of Life Sciences, Zhejiang University, Hangzhou 310027, China; 2College of Life Sciences-iCell Biotechnology Regenerative Biomedicine Laboratory, Joint Research Centre for Engineering Biology, Zhejiang University-University of Edinburgh Institute, Zhejiang University, Haining 314400, China; 3Key Laboratory of 3D Printing Process and Equipment of Zhejiang Province, School of Mechanical Engineering, Zhejiang University, Hangzhou 310027, China; 4Qiushi Academy for Advanced Studies, Zhejiang University, Hangzhou 310027, China; 5Key Laboratory of Biomedical Engineering of Ministry of Education, Zhejiang University, Hangzhou 310027, China; 6Zhejiang Provincial Key Laboratory of Cardio-Cerebral Vascular Detection Technology and Medicinal Effectiveness Appraisal, Zhejiang University, Hangzhou 310027, China

**Keywords:** human amniotic epithelial cell, neural cell, biomimetic scaffold, spinal cord injury

## Abstract

Spinal cord injury (SCI) results in devastating consequences for the motor and sensory function of patients due to neuronal loss and disrupted neural circuits, confronting poor prognosis and lack of effective therapies. A new therapeutic strategy is urgently required. Here, human amniotic epithelial cells (hAEC), featured with immunocompatibility, non-tumorgenicity and no ethical issues, were induced into neural-like cells by a compound cocktail, as evidenced with morphological change and the expression of neural cell markers. Interestingly, the hAEC-neural-like cells maintain the characteristic of low immunogenicity as hAEC. Aiming at SCI treatment in vivo, we constructed a 3D-printed GelMA hydrogel biomimetic spinal cord scaffold with micro-channels, in which hAEC-neural-like cells were well-induced and grown. In a rat full transection SCI model, hAEC-neural-like cell scaffolds that were implanted in the lesion demonstrated significant therapeutic effects; the neural circuit and hindlimb locomotion were partly recovered compared to little affection in the SCI rats receiving an empty scaffold or a sham implantation operation. Thus, the establishment of hAEC-neural-like cell biomimetic scaffolds may provide a safe and effective treatment strategy for SCI.

## 1. Introduction

As one of the major central nervous system diseases, spinal cord injury (SCI) results in devastating consequences in patients, not only permanent motor and sensory dysfunction, but also severe psychological issues as well [1,2,3,4]. However, current interventions for acute SCI are limited to surgical decompression and high-dose methylprednisolone administration, with limited effects on SCI repair [4,5]. A cascade of molecular and cellular events caused by the injury occur and evolve over days to months following the initial trauma, including neural cell death, inflammatory reactivity and cavity formation [6,7,8,9,10]. Given the poor regeneration of the central nervous system in the mammalian adult [11], replenishing neural cells and functioning in vivo safely and effectively has been the central goal of studies in regenerative medicine, which is also the basic requirement in clinical translation.

In recent years, numerous cellular therapies based on biomaterials have been developed for SCI repair, and some have obtained promising results in pre-clinical tests [12]. Among these studies, induced pluripotent stem cells (iPSCs) and neural stem cells (NSCs) are the major foci of researchers [13,14,15,16,17]. However, the concerns of safety (tumorigenicity and immunogenicity) and ethics hinder clinical applications [18,19]. As a type of perinatal cell with low immunogenicity, human amniotic epithelial cells (hAECs) do not express telomerase and HLA II antigens [20] but rather express non-classical HLA I antigens [21]. Therefore, hAECs exhibit unique advantage of biosafety (non-tumorigenesis and low immunogenicity) with no ethical concerns [22]. Moreover, hAECs express several pluripotent and neural markers and are able to secrete neurotrophic factors such as NGF and BDNF. Thus, hAECs may serve as an ideal seed cell for neural regeneration in SCI treatment [23,24,25,26,27,28].

Biomaterials are the backbones of engineered tissue scaffolds, supporting cell viability and promoting the therapeutic effects of cell grafts for SCI repair [29,30]. In tissue engineering, bio-scaffolds not only confer the damaged tissue with mechanical support but also deliver cell grafts, growth factors and anti-inflammatory drugs to the lesions [10,29]. Up until now, many kinds of hydrogels have been applied to address this difficult problem due to the advantages of biocompatibility, biodegradability, eliminating inflammation and promoting regeneration. Gelatin–methacryloyl (GelMA), as a kind of photocurable hydrogel material with good biocompatibility, has been widely used in the repair and regeneration of various tissues with few adverse reactions [31,32,33,34]. Combined with digital light processing (DLP) printing, GelMA is suitable for the personalized mass-fabrication of bio-scaffolds with complex architecture, facilitating practical applications in preclinical and clinical settings [35].

In the present study, we developed an inducing-treatment approach in vitro by a compound cocktail to induce hAEC into hAEC-neural-like cells, which were employed to improve the SCI carried by a biomimetic spinal cord scaffold composed of GelMA hydrogel by 3D printing.

## 2. Materials and Methods

### 2.1. hAECs Preparation

The samples of human placentas were obtained from healthy puerpera with written informed consent after a cesarean section. hAECs were prepared as described previously [22]. The procedure was approved by the Institutional Patients and Ethics Committee of the International Peace Maternity and Child Health Hospital, Shanghai Jiao Tong University School of Medicine. All puerpera carried out medical experiments to make sure to be negative for hepatitis, HIV and treponema pallidum. The hAECs used in this research were provided by Shanghai iCELL Biotechnology Co Ltd. (Shanghai, China). For hAECs isolation, we isolated and washed amniotic membrane in PBS buffer times, then digested it with 0.25% trypsin for 20 min in a 37 °C water bath. Later, the suspension was centrifuged for 8 min at 500× *g* to collect hAECs. Isolated hAECs were cultured in complete culture medium F12/DMEM with 2 mM L-glutamine, 1% sodium pyruvate, 1% non-essential amino acid, 1% antibiotic–antimycotic and 10% fetal bovine serum (Thermo Fisher Scientific, Grand Island, NY, USA) and 10 ng/mL human EGF (Peprotech, Cranbury, NJ, USA) in the incubator with 5.5% CO_2_ at 37 °C for a few days until hAECs attached and filled the dish. 

### 2.2. Establishment of BGVB Inducement Approach

For ease of clarity, we named the inducement approach as BGVB based on the initials abbreviation of the major components of the inducing system (BDNF, GDNF, Vitamin C, B27 supplement). The BGVB-induced medium was Neurobasal Medium (Gibco, Grand Island, NY, USA) with 20 ng/mL BDNF (Peprotech, Cranbury, NJ, USA), 20 ng/mL GDNF (Peprotech, Cranbury, NJ, USA), 200 μM Vitamin C (Sigma, Madison, WI, USA) and B27 Supplement (Gibco, Grand Island, NY, USA), and with the same concentration of 4 supplements as 2.1 (L-glutamine, non-essential amino acid, sodium pyruvate and antibiotic–antimycotic). hAECs were seeded in the 6-well plate and cultured for 7 days with BGVB-induced medium. Then, the expression of neural markers was measured by q-PCR and immunofluorescence staining, and epithelial and mesenchymal markers were detected by flow cytometry.

### 2.3. GelMA Preparation and Biomimetic Spinal Cord Scaffold Printing

GelMA is synthesized by gelatin (Aladdin, Shanghai, China) and methacrylate (Aladdin, Shanghai, China) in phosphate buffer (pH 7.4). Methacrylate (MA) was added into the solution slowly to prevent solidification. The temperature was kept around 50 °C to avoid decomposition of reactants while maintaining a high reaction rate. For the same purpose, the pH was controlled between pH 8 and 9 by adding sodium hydroxide solution. The reaction was completed when the pH no longer changed. Then the mixture was diluted to 5 times volume by pure water and then was transferred to the dialysis unit. During dialysis, unreacted MA and side reaction products were fully removed, and the process was stopped automatically until an ion sensor indicated that the products were qualified. Finally, the products were freeze-dried to remove water and stored at –20 °C.

The DLP bioprinting system was composed of three parts: optical units, material storage units and mechanical movement units. GelMA with photo-crosslinking properties was generated by dissolving the GelMA into PBS buffer for making the 15% (*w*/*v*) solution followed by adding 0.25% (*w*/*v*) initiator lithium acylphosphinate (LAP). The liquid GelMA was then filled into the material storage unit. According to designed structures that projected to the printing platform, the scaffolds were generated by a 3D printer using the liquid GelMA. Finally, the printed scaffolds were immersed into 75% ethanol and PBS buffer, respectively, for further experiments.

### 2.4. Flow Cytometry

To evaluate the characteristics and purity of hAECs, we employed flow cytometry to detect the expression of relative cell markers. E-cadherin (Biolegend, San Diego, CA, USA) and SSEA4 (Biolegend, San Diego, CA, USA) were used to test the epithelial properties and plasticity of hAECs, respectively. CD34 (Invitrogen, Grand Island, NY, USA), CD45 (Biolegend, San Diego, CA, USA), CD31 (Invitrogen, Grand Island, NY, USA), CD144 (Invitrogen, Grand Island, NY, USA), CD73 (Invitrogen, Grand Island, NY, USA) and CD90 (Invitrogen, Grand Island, NY, USA) were used to evaluate the purity and basic characteristics of hAECs. To indicate the immunogenicity of hAECs before and after inducing treatment, HLA-DR (Biolegend, San Diego, CA, USA), HLA-DQ (Biolegend, San Diego, CA, USA) and HLA-G (Biolegend, San Diego, CA, USA) were stained after cells incubated with 10 ng/mL IFN-γ (Peprotech, Cranbury, NJ, USA) for 3 days. All procedures followed the manufacturers’ instructions, and later, analysis was conducted with flow cytometry (Beckman, Brea, CA, USA). Analyses were performed on five biological replicates.

### 2.5. Enzyme-Linked Immunosorbent Assay

To examine the secretion of neurotrophic factors, hAECs were cultured in 10 cm dishes for 3 days, and the suspension was collected for Enzyme-Linked Immunosorbent Assay (ELISA). The operating protocol followed the standard procedures provided by the nerve growth factor (NGF) and brain-derived neurotrophic factor (BDNF) ELISA Kits (R&D Systems, Minneapolis, MN, USA). 

### 2.6. Quantitative Real-Time PCR

To test the expression of neural genes of hAECs before and after inducing treatment, the total RNA was extracted with the SteadyPure Universal RNA Extraction Kit (Accurate Biology, Changsha, China). Then the reverse transcription assay was performed followed by the procedures of the Evo M-MLV RT Premix Kit (Accurate Biology, Changsha, China). Later, quantitative Real-Time PCR was performed based on the manufacturer’s instructions of SYBR Green Premix Pro Taq HS qPCR Kit (Accurate Biology, Changsha, China). The primers used for the qPCR are listed in the Appendix A. Human glyceraldehyde 3-phosphate dehydrogenase (GAPDH) was used as the control, and all tests were performed on 3 replicates. 

### 2.7. Pre-Labelling of hAECs

To trace the hAEC-neural-like cells in vivo, we pre-labeled hAECs with CellTracker CM-Dil (Invitrogen, Grand Island, NY, USA) in vitro before inducing treatment. Briefly, hAECs were incubated in the basic DMEM media with 2 μg/mL CM-Dil for 5–8 min in 37 °C and then transferred into 4 °C for 15 min. After that, hAECs was seeded into dishes and cultured for 2 days. 

### 2.8. hAECs Seeding and hAEC-Neural-like Cells Induction on Biomimetic Spinal Cord Scaffold

First, 1 × 10^6^ hAECs were resuspended in 100 μL culture medium. To seed cells into the micro-channels, hAEC suspensions were seeded using cuspidal pipette tips or 1 mL syringe needles. To avoid the influence of the single direction deposition from gravity, we seeded the cell suspension several times and put the scaffold in different directions, so that the seeded hAECs could deposit in as many surfaces as possible. The scaffolds were let to stand for about 1 h in a cell incubator to enable hAECs to attach better on surfaces. Then the cell-loaded scaffolds were cultured with BGVB-induced medium for 5 days before further experiments.

### 2.9. Scanning Electron Microscopy and Imaging

The study used SEM to observe the surface morphology of biomimetic scaffolds and hAECs seeded on the scaffolds. GelMA scaffold samples were immersed into 2.5% glutaraldehyde for more than 24 h, and then they were fixed in 1% osmium tetroxide for 2 h. After dehydrating in gradient ethanol (30%, 50%, 70%, 80%, 90%, 95%, 100%), samples were dried with a supercritical point dryer (HITACHI, Tokyo, Japan) and sputter-coated. After that, the biomimetic scaffold samples were captured using scanning electron microscopy (HITACHI, Tokyo, Japan) at the Analysis Center of Agrobiology and Environmental Science, Zhejiang University.

### 2.10. Animals

Female SD rats of 230–250g were obtained from Slac Laboratory Animal Co. Ltd. (Shanghai, China). All rats were randomly divided into 4 groups: normal rats (n = 5), full transection control rats (n = 5), rats receiving empty scaffold (n = 5) and rats receiving biomimetic scaffolds loaded with hAEC-neural-like cells (n = 5). All assessments of behavioral and histological tests were performed using images blinded to the observed. All animal experimental protocols were approved by the Laboratory Animal Welfare and Ethics Committee of Zhejiang University (Ethics Code: ZJU20210074).

### 2.11. T10 Full Transection Injury and Post-Surgical Care

The operation of laminectomy and T10 transection was similar to that described elsewhere [36,37]. Briefly, rats were deeply anesthetized, and local hair was cleaned to expose the skin over T10. After that, a midline incision was made by scalpel over T10 to expose the vertebral plate. Then a T9–T10 laminectomy was implemented on rats to expose spinal cord tissue. With the micro-scissors and micro-stereoscope, a transection surgery of the T10 spinal cord was implemented carefully. After hemostasis, a 2 mm spinal cord block was replaced with a 2.5–3 mm empty scaffold, the biomimetic scaffold loaded with hAEC-neural-like cells or 100 μL PBS solution. Following implantation, the dorsal muscles and skin were sutured, and antibiotics were administered. After operation, 1 mL saline was injected in rates subcutaneously, and rats were kept in a warm room until awakening. Then we manually emptied rats’ bladders twice a day and injected Baytril (1 mg/100 g) for 5 days to prevent urinary infection [38].

### 2.12. EMG Recording and Cortical Stimulation

For MEP recording by cortical stimulation, the rats were deeply anesthetized, and the skulls were exposed. The stimulation position of customized stimulation electrodes was in the motor cortex of the left cerebral hemisphere, with the accurate position (x, y, z) being (2.95 mm, 1.65 mm, 2 mm) relative to Bregma (0, 0, 0). Recording electrodes were wrapped to the mid-belly of the right TA muscle, with the other end connecting with the OmniPlex/128 Plexon (Plexon, Dallas, TX, USA). The single electrical stimuli (5 mA constant current stimulus, 10 ms duration and 5 s intervals) was generated by a pulse generator and isolator (A–M System, Olympic Peninsula, WA, USA). EMG recordings were recorded and examined by a technician blinded to the treatment groups.

### 2.13. Behavioral Assessments

The Basso, Beattie and Bresnahan (BBB) locomotion score in open-field was examined weekly by investigators blinded to the treatment groups over an 8-week period post implantation [39]. The rats were free to crawl in a 2 m^2^ area, and hindlimb locomotion was assessed and recorded for a 4-min period. Before assessment, researchers needed to empty rats’ bladders manually to avoid the influence of moving.

### 2.14. Immunofluorescence and Imaging

For immunofluorescence of the spinal cord tissues, samples were fixed with paraformaldehyde (PFA) for 24 h and cryo-protected with 30% sucrose at 4 °C for 3 days. Then tissues were embedded with O.C.T. compound (4583, Sakura, Japan) in a cryostat and sectioned in 30 μm sections. The sections were treated with 0.5% Triton-X-100 for 2 h, followed by a blocking solution containing 5% normal donkey serum and 5% bovine serum albumin at room temperature. Then the sections were incubated with primary antibodies at 4 °C overnight: Anti-TUJ1 antibody (Abcam, Cambridge, UK); Anti-GAP43 antibody (Millipore, Boston, MA, USA); and Anti-NF-H antibody (Abcam, Cambridge, UK). 

For immunofluorescence of cells, samples were fixed with paraformaldehyde (PFA) for 30 min and treated with 0.5% Triton-X-100 for 5 min, followed by a blocking solution containing 5% normal donkey serum and 5% bovine serum albumin at room temperature for 1 h. Then samples were incubated with primary antibodies at 4 °C overnight: Anti-NeuN antibody (Abcam, Cambridge, UK); Anti-NF-H antibody (Abcam, Cambridge, UK); and Anti-MAP2 antibody (Proteintech, Wuhan, China).

Secondary antibodies (room temperature for 1 h) included: donkey anti-mouse/rabbit IgG secondary antibody, Alexa Fluor 488 (Invitrogen, Grand Island, NY, USA); and donkey anti-mouse/rabbit IgG secondary antibody, Alexa Fluor 594 (Invitrogen, Grand Island, NY, USA). Spinal cord section scans were imaged with an Olympus VS200, and other transverse and horizontal sections were imaged with confocal microscopes FV3000 (Olympus, Tokyo, Japan).

### 2.15. Quantification and Statistical Analysis

Statistical analysis was performed in GraphPad Prism 9. One-way and two-way ANOVA with Tukey’s and Bonferroni multiple comparisons tests were used to analyze the significance of more than two samples. Data were presented as means ± SEM. *p* values < 0.05 were considered to be statistically significant.

## 3. Results

### 3.1. The Basic Characteristics of Human Amniotic Epithelial Cells

To guarantee the quality of hAECs, we first tested the epithelial characteristics of isolated hAECs. The morphology of hAECs showed the typical epithelial appearance, with close arrangement and clear cell borders (Figure 1a). The representative epithelial markers, cytokeratin and E-cadherin, were positive in primary hAECs (Figure 1b,c), which was consistent with the results of flow cytometry (Figure 1d). Moreover, we also detected the purity of hAECs. The isolated hAECs were negative to hematopoietic lineage markers CD34 and CD45 (Figure 1e,f) and endothelial marker CD31 and CD144 (Figure 1g,h). In addition, the flow cytometry results also demonstrated that hAECs were positive to mesenchymal markers CD73 and CD90 (Figure 1i,j). These data showed that the hAECs cultured in our procedure possessed typical epithelial properties with decent cell purity. We further identified that hAECs expressed some pluripotent markers such as NANOG, OCT4 and SSEA4 by immunofluorescence and flow cytometry (Figure 1k–n), proving the multidirectional differentiation potential of hAECs, as suggested in previous studies [23,25,28]. Additionally, the immunofluorescence staining results showed the expression of neural stem cell markers including VIMENTIN and NSE in hAECs (Figure 1o,p), indicating their neural plasticity. Consistently, ELISA assays demonstrated that the culture supernatant of hAECs contained a certain concentration of NGF and BDNF (Figure 1q). These results above indicated a well-established hAEC culture system and their specific neural characteristic and plasticity, which paved the way for further experiments. 

### 3.2. The Establishment of BGVB Inducing-Treatment Approach

To obtain hAEC-derived neural cells for cell replenishment in SCI lesions, we tried to establish an inducing-treatment approach by screening several small molecules and neurotrophic factors. Five different approaches were designed to culture hAECs in vitro (Appendix A). Basically, hAECs cultured with method 4 maintained high viability and displayed neuron-like morphology (Appendix A). The expression of representative neural markers was assessed by quantitative real-time PCR analysis, which showed that the neural markers of cells treated with method 4 and 5 were induced since day 4 (Appendix A). Furthermore, the expression level of these genes was significantly enhanced in these two groups of cells when cultured for 7 days, especially the early nerve development and neural stem cell-related genes such as Pax6, Sox2 and Nkx6.1 (Appendix A). Given the obvious apoptosis in method 5 after passaging, which suggested the cytotoxicity of the treatment system, we fixed method 4 as the inducing-treatment approach suitable for hAECs. 

According to the screening results above, BDNF, GDNF, Vitamin C and B27 supplement (named as BGVB based on the initials abbreviation of the four major components) in method 4 conferred hAECs with neural characteristics to generate hAEC-neural-like cells (Figure 2a). Compared to the epithelial morphology (Figure 2b), hAEC-neural-like cells displayed more obvious nuclei and diverged structures (Figure 2c). After passaging, significant neuron-like appearances and cell structures were observed (Figure 2d), suggesting that properties of hAECs were switched by BGVB treatment. Indeed, quantitative real-time PCR analysis identified the enhancement of a series of signature neural markers including NeuroD1, Map2 and NeuN in hAEC-neural-like cells (Figure 2e). The immunofluorescence assays showed that the induced cells were positive to TUJ1, and those cells with neuronal morphology expressed mature neuron markers NH-F and MAP2 (Figure 2f,g). Moreover, the hAEC-neural-like cells expressed mature neuron nucleus protein NeuN (Figure 2h). In addition to the evaluation of neural properties, flow cytometry results also indicated that the epithelial characteristic of hAEC-neural-like cells was enhanced by BGVB inducement (Appendix A), along with the reduced expression of mesenchymal markers such as CD73 and CD90 (Appendix A). These data demonstrated that BGVB treatment could induce hAECs into neural-like cells significantly, as evidenced by both morphological change and neural marker expression. 

In addition, we also examined the immunogenicity of hAEC-neural-like cells. The results of flow cytometry showed the negative expression of HLA II antigen in both non-treated hAECs and hAEC-neural-like cells in the presence of IFN-γ (Figure 3a–d). Notably, the expression of HLA-G was significantly increased in hAEC-neural-like cells after BGVB treatment, suggesting that hAEC-neural-like cells maintain the advantage of low immunogenicity (Figure 3e,f). 

### 3.3. The Establishment of 3D Biomimetic Spinal Cord Scaffold Loaded with hAEC-Neural-like Cells

Before manufacturing, we optimized the GelMA substitution degree suitable for attachment and growth of hAECs based on the mechanical properties of the hydrogel influence cell state and behaviors. hAECs were seeded on GelMA surfaces with four different substitution degrees to test their attachment and growth. The results indicated that the GM-100 GelMA surface performed a better biocompatibility for hAECs, as evidenced by the large number of cells (Appendix A) and the significant morphological changes (Appendix A). Consequently, we chose the GM-100 GelMA hydrogel for further experiments. 

Aiming at SCI treatment in vivo, we designed and manufactured the GelMA hydrogel scaffold with micro-channels by 3D printing to mimic the structure and size of rat spinal cord (Figure 4a). The overall dimensions of the scaffold products were 2.5 mm × 2 mm × 2 mm, with nine inner channels of 150 μm diameter (Figure 4b). In order to achieve the large and uniform cell loading, we seeded hAECs on scaffolds several times in different directions (see details in Section 2.4) followed by BGVB treatment in situ for 7 days. Immunofluorescence staining showed high expression of the neuron-specific markers MAP2, NF-H, TUJ1 and NeuN after three-dimensional BGVB treatment for 7 days, indicating the successful induction of hAECs to hAEC-neural-like cells in the 3D scaffold (Figure 4c–f). The live/dead assay demonstrated viability maintenance of most hAEC-neural-like cells, suggesting that the scaffold provided hAEC-neural-like cells with suitable micro-environments for induction and growth (Figure 4g,h). Moreover, cell skeleton staining showed that most cells were concentrated in the inner channels with a small number of cells present on the surface (Figure 4i). Notably, the magnified image confirmed the morphological feature of the loaded hAEC-neural-like cells (Figure 4j), which was similar to the appearance of hAEC-neural-like cells obtained in dish. Vertical observation of the inner channels demonstrated that the walls of the channels were clear (Figure 4k), and cells were distributed in the whole channel (Figure 4l), confirming the good biocompatibility of the scaffolds. Additionally, the scanning electron microscopy (SEM) results showed the whole appearance of the scaffold with inner channels (Figure 4m), and magnified images further indicated the robust and abundant attachment of induced hAEC-neural-like cells (Figure 4n,o). Moreover, the neural-like network was formed on the scaffold surface (Figure 4p,q), which was similar to that cultured in the dish.

### 3.4. The Biomimetic Scaffold Loaded with hAEC-Neural-like Cells Replenished Neural Cells and Improved Locomotion in SCI Rat Model

To examine the therapeutic effects of the biomimetic scaffold-based hAEC-neural-like cells on SCI in vivo, the scaffolds loaded with hAEC-neural-like cells were implanted into rats with full transection SCI. Considering the protection of scaffolds to cell grafts in the hostile pathological milieus [15], we transplanted the biomimetic scaffolds in the acute SCI phase. In detail, a T9–T10 laminectomy was performed to expose spinal cord tissue (Figure 5a). Then, full transection operation of T10 was performed carefully to remove a block of 2 mm tissue (Figure 5b). After successful hemostasis, the scaffolds loaded with hAEC-neural-like cells or empty scaffolds were implanted into the lesions of SCI rats (Figure 5c). To determine the locomotion recovery, the hindlimb locomotion of all SCI rats from the indicated groups were examined weekly over an 8-week period. Basso, Beattie and Bresnahan (BBB) scores indicated that the locomotion of rats receiving scaffolds loaded with hAEC-neural-like cells was improved noticeably in 6 weeks compared with rats with empty scaffolds. Eight weeks later, the BBB scores reached a mean value of 5.667 in SCI rats implanted with hAEC-neural-like cell scaffolds, indicating movement about each joint of the hindlimb, compared to a mean score of 2.333 points in rats with empty scaffolds, and 0.667 points in full transection SCI rats with sham operation of scaffold implantation (FT control) (Figure 5d). We further investigated the innervation capacity of brain to hindlimbs by measuring motor-evoked potential (MEP). At 8 weeks post-implantation, rats receiving hAEC-neural-like cell scaffolds exhibited strong responses of MEP (mean MEP amplitude reached 29.57% of that in normal rats) compared to lower readings in rats with empty scaffold and FT control rats (Figure 5e,f).

On the other hand, histological analysis of the SCI lesions from the indicated groups was performed based on the locomotion testing results. The immunofluorescence staining 8 weeks post-implantation demonstrated a large number of NF-H marked neural axons regenerated in biomimetic scaffolds loaded with hAEC-neural-like cells. Robust axon projections and growth along the channels was observed in lesions. In contrast, no obvious NF-H signal was detected in lesion of FT control rats, indicating the formation of a non-neural lesion core [9]. A few NF-H marked axons were observed around the lesion border in rats with empty scaffolds, but no axon was found projected and growing into the lesions (Figure 6a–c). Further examination of axonal development in scaffold channels also indicated the continuous and robust projection of axons in rats implanted with hAEC-neural-like cells scaffolds, while scattered and weak signals were observed in rats receiving empty scaffolds (Figure 6d). Interestingly, part of the hAEC-neural-like cells were found colocalized with the NF-H marked neural axon in the scaffold, as evidenced by confocal microscopy (Figure 6e white arrow heads). In addition, the evaluation of the inflammation response showed that significant activation of macrophage and microglia was persistently present around lesions in FT control rats and empty scaffold rats (Appendix A), revealing the typical tissue micro-environment after SCI. However, no significant difference of inflammatory signals was observed between rats receiving biomimetic scaffolds loaded with hAEC-neural-like cells and other groups, which demonstrated that hAEC-neural-like cells had little effect on neuroinflammation (Appendix A). All these data above indicated that the hAEC-neural-like cell biomimetic scaffolds facilitated replenishment of neural cells in lesions and the consequent locomotor recovery in the SCI rat model. 

## 4. Discussion

Up until now, tissue engineering has been considered as one of the most promising strategies for the treatment of SCI, aiming to re-establish the damaged neural circuits and recover locomotion [40]. Given the loss of neurons and glial cells in SCI lesions, replenishment of neural cells could have important implications for SCI recovery [41]. In the current study, we explored the manufacture of a biomimetic spinal cord scaffold loaded with hAEC-derived neural-like cells for SCI repair. Based on their plasticity and neural characteristics, we induced hAECs into neural-like cells both in 2D culture and 3D biomimetic scaffolds by a specific BGVB compound cocktail, indicating an epithelial-to-neural trans-differentiation process. Importantly, implantation of the hAEC-neural-like cell scaffold improved locomotion of the hindlimbs of SCI rats, which was proven by robust axon projection and growth of axons in the lesions. These results indicated that the biomimetic spinal cord scaffolds had the potential to replenish neural cells in lesions and restore partial motor function of SCI rats.

For clinical translation of cell therapy, safety is both the bottom line and the top priority; the process must be controllable regarding the fate of implanted cells, it must not be tumorigenic and immune rejection must be minimized. Owing to the versatile differentiation capacity, pluripotent stem cells (ESCs and iPSCs) and NSCs are still the main cell types investigated in studies of SCI repair. However, concerns about ethics and safety of ESCs, as well as uncovered tumorgenicity and immunogenicity of iPSCs, have restrained their clinical applications. Furthermore, NSCs have confronted challenges of relatively low yield and susceptible differentiation destination. Several studies have reported that NSCs were prone to generating astrocytes in the condition of neuroinflammation after SCI, leading to the failure of regeneration [42,43]. In the present study, we focused on hAECs, which were identified with low immunogenicity and no tumorgenicity due to the defect in telomerase expression, as reported in our previous studies [21,22,26,27,44]. Intriguingly, the property of neural plasticity highlighted the potential of hAECs as seed cells for neural cell replenishment in SCI repair [45]. Indeed, one-step induction could achieve high-yield hAEC-derived neural-like cells, which demonstrated significant therapeutic effects on SCI in an animal disease model. Of note, the hAEC-neural-like cells maintained the advantages of low immunogenicity, even upon pro-inflammatory stimuli. In vivo, the observation of some pre-labeled hAEC-neural-like cells in the scaffold after implantation without any immunosuppressive agent administration may also suggest the immunological–adaption characteristic of hAEC-neural-like cells. Combined with the absence of ethical issues, hAECs may meet the requirements of clinical-grade seed cells for therapy of neural injuries.

Although obvious morphological changes and the expression of a series of neuron markers were detected, hAEC-neural-like cells are not comparable to the native neurons. In other words, hAEC-neural-like cells may have some characteristics of neural progenitors. Therefore, hAEC-neural-like cells may act as either replenished cells or neurotrophic cells for the neural circuit recovery in the implanted scaffolds. In terms of hAEC-to-neural cell induction, more detailed studies are still needed, especially in achieving matured neural cells efficiently. We will mainly focus on screening new compounds or chemicals for induction, both in 2D and 3D culture conditions. However, we anticipate that a large portion of the inducers will be in categories different from those for pluripotent stem cell differentiation, because generation of the hAEC-derived neural cells should be based on the mechanism of trans-differentiation. Some previous studies found bipotential precursors that could differentiate into both mesenchymal and endothelial cells [46,47,48]. Given the epithelial and mesenchymal properties of hAECs, coupled with the up-regulated expression of epithelial and neural markers after BGVB treatment, we speculate that BGVB induced the characteristic switch from hAECs to hAEC-neural-like cells in a mesenchymal–epithelial transition (MET)-like manner, inspired by the differentiation of brain vascular pericytes after ischemia [49]. In other words, hAEC-neural-like cells could be regarded as neuroepithelial cells with powerful neural properties and promising application scenarios, while the more detailed mechanism must still be investigated in the future. In vivo results demonstrated the significant improvement of locomotion and histology; however, some issues still need to be addressed, such as severe neuroinflammation post-surgery, and the low number of animals (N = 3) in these analyses due to a number of deaths after operation. Our future work will focus on both replenishing neural cells and inhibiting neuroinflammation after SCI to obtain better therapeutic effects with updated bioengineering strategies. Furthermore, it is necessary to improve the surgery operation and prognosis to increase sample size, in order to achieve surgeries with these promising results.

In the present study, a biomimetic scaffold using GelMA hydrogel was constructed for carrying the hAEC-neural-like cells for repairing SCI in vivo. The scaffold was manufactured with micro-channels to mimic the size and shape of the rat spinal cord. On the other hand, the excellent biocompatibility of the scaffold for hosting hAEC-derived neural-like cells and the neural axon growth was further identified in our study. The complex structures with micro-channels not only contribute to the axon projection and neural circuits connection but also exert protective effects on grafted cells. The establishment of biomimetic scaffolds may expand the application range of hAEC-derived cells in cell therapy. From the perspective of manufacturing, it is required to develop bio-scaffolds with high cell-loading and capacity of active cell-absorption for simplifying the tedious procedures of cell seeding in the future. Improved therapeutic efficiency and convenient procedures are crucial for clinical trials.

## 5. Conclusions

In conclusion, our study suggests an alternative bioengineering system for regenerative medicine of SCI. We provide an efficient strategy to obtain high yield neural-like cells with reliable biosafety. Aiming at SCI treatment in vivo, a 3D-printed biomimetic scaffold with excellent biocompatibility and a complex structure was manufactured using GelMA hydrogel. The 3D cell therapy system based on hAEC-neural-like cell bio-scaffolds demonstrated significant therapeutic effects in a rat SCI model, indicating its potential application in the clinic.

## Figures and Tables

**Figure 1 bioengineering-09-00535-f001:**
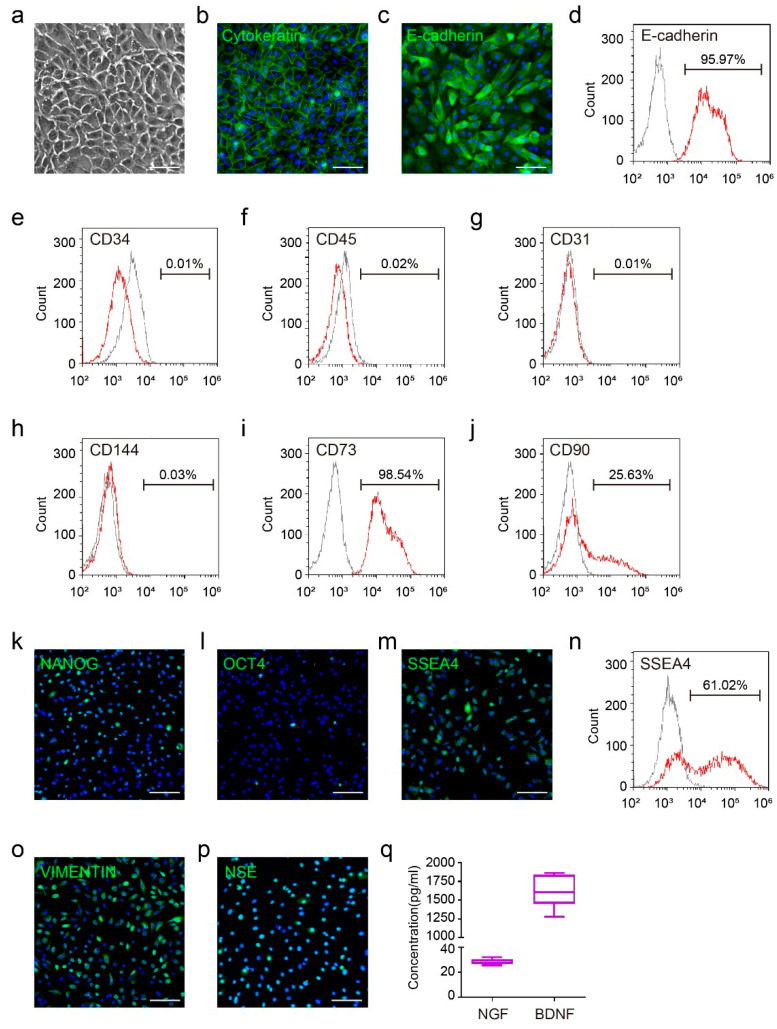
The basic characteristics of primary hAECs. (**a**) The morphology of primary hAECs; (**b**,**c**) the immunofluorescence staining of epithelial marker Cytokeratin and E-cadherin of primary hAECs; (**d**) the flow cytometry results of epithelial marker E-cadherin of primary hAECs; (**e**–**j**) the flow cytometry results of hematopoietic markers CD34 and CD45, endothelial markers CD31 and CD144 and mesenchymal markers CD73 and CD90 of primary hAECs; (**k**–**m**) the immunofluorescence staining of pluripotent markers NANOG, OCT4 and SSEA4 of primary hAECs; (**n**) the flow cytometry results of pluripotent marker SSEA4 of primary hAECs; (**o**,**p**) the immunofluorescence staining of neural markers VIMENTIN and NSE of primary hAECs; (**q**) the ELISA results of neurotrophic factors NGF and BDNF of primary hAEC culture supernatant, n = 3. Scale bars: (**a**–**c**): 50 μm; (**k**–**m**): 50 μm; (**o**,**p**): 50 μm.

**Figure 2 bioengineering-09-00535-f002:**
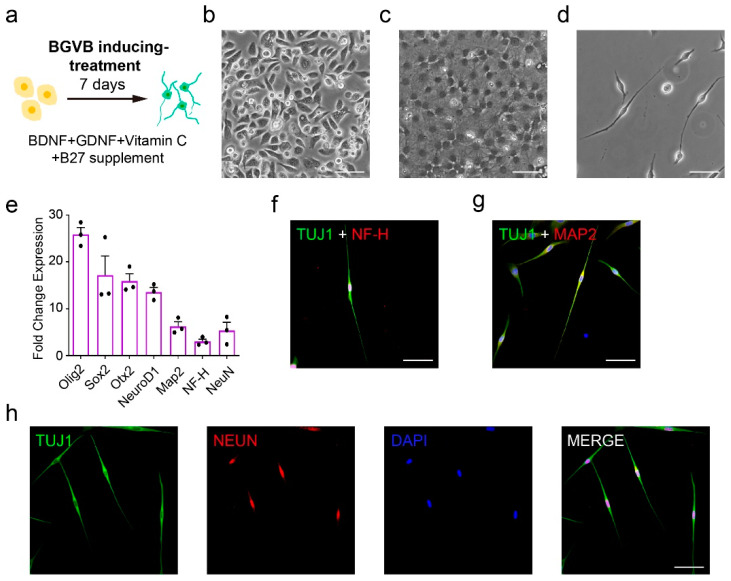
The neural characteristic examination of hAEC-neural-like cells induced by the BGVB approach. (**a**) Experimental schematic of the BGVB inducing-treatment approach; (**b**) the morphology of non-treated hAECs; (**c**) the morphology of hAECs after BGVB treatment for 4 days; (**d**) the morphology of BGVB-treated hAECs 3 days after passaging; (**e**) the activation of neural markers after a 7-day period of BGVB inducement, normalizing the non-treated hAECs as 1, n = 3; (**f**,**g**) the immunofluorescence staining of neural markers TUJ1, NF-H and MAP2 after BGVB inducement; (**h**) the immunofluorescence staining of mature neuron nucleus protein NeuN after BGVB treatment; (**i**) the flow cytometry results of HLA antigens of hAECs before and after treatment with the BGVB approach. Scale bars: (**b**–**d**): 50 μm; (**f**,**g**): 50 μm; (**h**): 50 μm.

**Figure 3 bioengineering-09-00535-f003:**
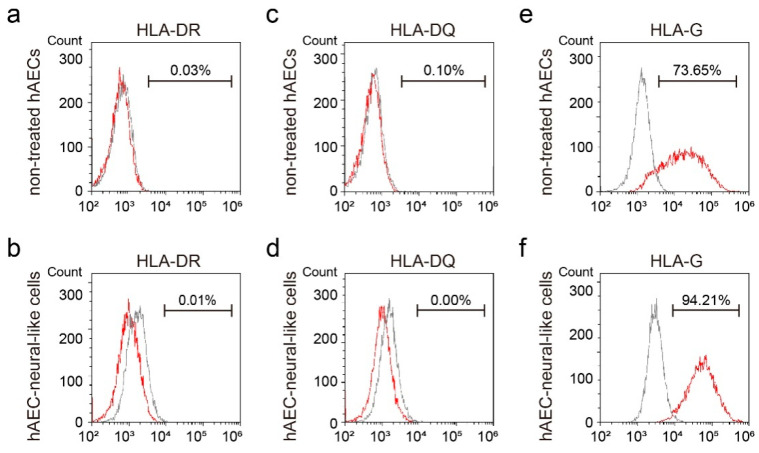
The immunogenic characteristics of hAEC-neural-like cells induced by the BGVB approach. (**a**–**d**) The flow cytometry results of HLA II antigens of hAECs and hAEC-neural-like cells. (**e**,**f**) The flow cytometry results of non-classical HLA I antigens (HLA-G) of hAECs and hAEC-neural-like cells.

**Figure 4 bioengineering-09-00535-f004:**
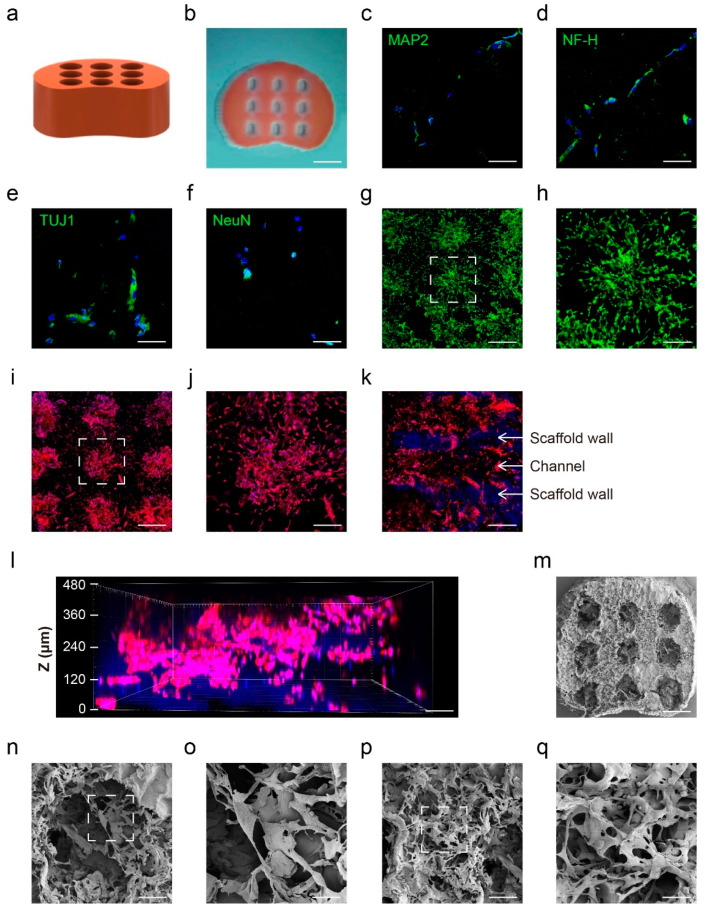
The in vitro performance of hAEC-neural-like cell biomimetic scaffolds. (**a**,**b**) The design and real product of biomimetic rat spinal cord scaffold; (**c**–**f**) the immunofluorescence staining of neuron specific markers MAP2, NF-H, TUJ1 and NeuN to the hAEC-neural-like cell biomimetic scaffolds; (**g**,**h**) the live/dead staining of hAEC-neural-like cells on scaffolds, green fluorescence-represented live cells and red fluorescence-represented dead cells; h is the magnified image of the white dotted frame in g; (**i**,**j**) the cell skeleton staining of hAEC-neural-like cells on scaffolds observed from the radial direction, red fluorescence-represented cell skeleton, blue fluorescence-represented cell nucleus; j is the magnified image of the white dotted frame in i; (**k**) the cell skeleton staining of hAEC-neural-like cells on scaffolds observed from the axial direction, red fluorescence showed the distribution of cells in channels, blue fluorescence represented the autofluorescence of scaffold walls; (**l**) 3D view of cells distribution in the scaffold channel, red fluorescence represents the cell skeleton, blue fluorescence represents the cell nucleus; (**m**–**q**) SEM images of hAEC-neural-like cells on scaffolds; (**m**) SEM image of the whole biomimetic scaffold loaded with hAEC-neural-like cells; (**n**) SEM scanning of morphology of cells in micro-channel; (**o**) the magnified image of the white dotted frame in n; (**p**) the cell network formed by hAEC-neural-like cells on the scaffold surface; (**q**) the magnified image of the white dotted frame in p. Scale bars: (**b**): 700 μm; (**c**–**f**): 100 μm; (**g**): 650 μm; (**h**): 250 μm; (**i**): 650 μm; (**j**): 250 μm; (**k**): 500 μm; (**l**): 80 μm; (**m**): 400 μm; (**n**): 150 μm; (**o**): 35 μm; (**p**): 150 μm; (**q**): 50 μm.

**Figure 5 bioengineering-09-00535-f005:**
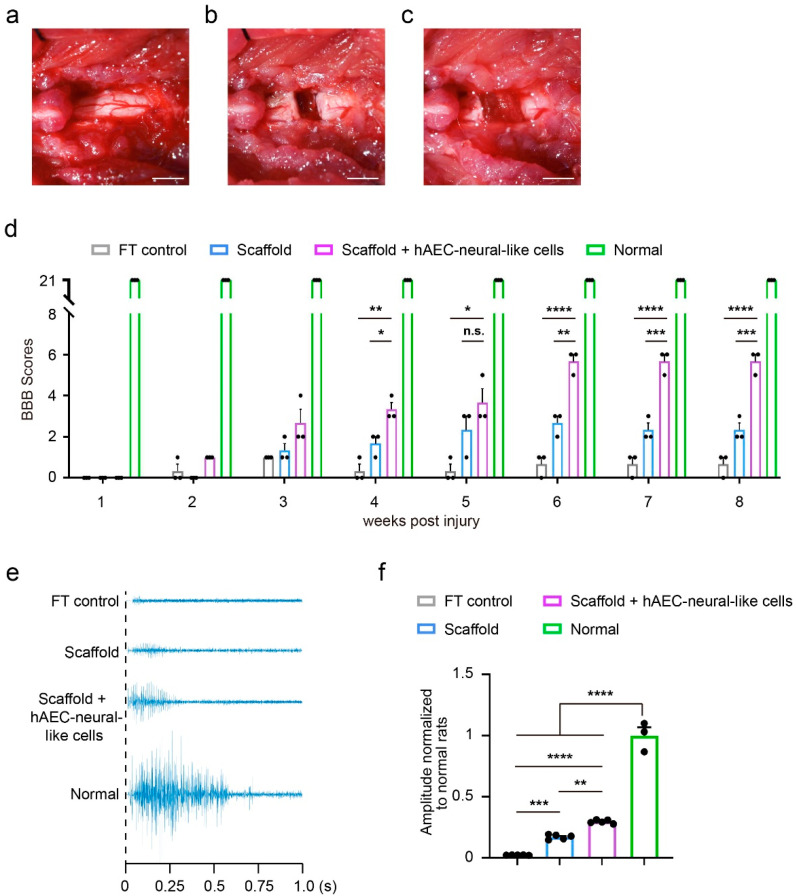
Implantation of hAEC-neural-like cell biomimetic scaffolds in the lesion partially rescue hindlimb locomotion in a rat model of full transection spinal cord injury. (**a**–**c**) The operation procedure. (**a**) Exposure of spinal cord tissue after laminectomy; (**b**) full transection operation of T10; (**c**) implantation of hAEC-neural-like cell biomimetic scaffolds into the lesion; (**d**) BBB locomotion scores within 8 weeks, one-way ANOVA followed by Bonferroni’s multiple comparisons test, n = 3 per group, **** *p* < 0.0001, *** *p* < 0.001, ** *p* < 0.01, * *p* < 0.05, n.s. not significant, error bars, SEM (the BBB scores were measured repeatedly in the same rat along 8 weeks post implantation, and the scores of rats among different groups at the same time point were analyzed to determine the statistical difference between two groups); (**e**) representative responses in the TA muscle evoked by epidural motor cortex stimulation in full transection control, empty scaffold (scaffold), hAEC-neural-like cells biomimetic scaffolds (scaffold + hAEC-neural-like cells) and normal rats; (**f**) quantitative analysis of the MEP response amplitude from indicated groups, One-way ANOVA, followed by Tukey’s multiple comparisons test, n = 5 for FT, scaffold and scaffold + hAEC-neural-like cells groups; n = 3 for normal group, **** *p* < 0.0001, *** *p* < 0.001, ** *p* < 0.005, error bars, SEM.

**Figure 6 bioengineering-09-00535-f006:**
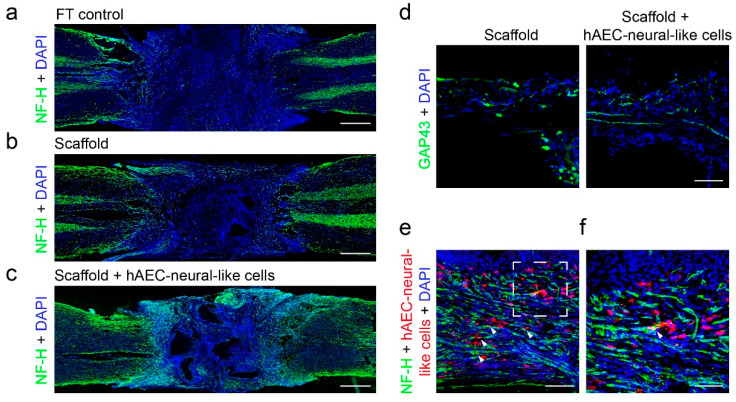
Implantation of hAEC-neural-like cell biomimetic scaffolds in the lesion partially recover the neural circuit in a rat model of full transection spinal cord injury. (**a**–**c**) The fluorescence images of NF-H (green) axons showing the growth of nerve fibers around the lesion from the indicated groups; (**d**) the growth and projection of axons in micro-channels between empty scaffold and hAEC-neural-like cell biomimetic scaffold groups at 8 weeks post implant; (**e**,**f**) the fluorescence images indicated that some of the NF-H-marked axons co-localized with hAEC-neural-like cells (pre-labeled with red fluorescence) in channels (white arrow heads), f is the magnified image of the white dotted frame in e. Scale bars: (**a**–**c**): 1 mm; (**d**): 100 μm; (**e**): 120 μm; (**f**): 60 μm.

## Data Availability

All data are included in the text and Appendix A.

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
