# Peer review of "Therapeutic Effect of Biomimetic Scaffold Loaded with Human Amniotic Epithelial Cell-Derived Neural-like Cells for Spinal Cord Injury"

_bioengineering, 2022, doi:10.3390/bioengineering9100535_

Round 1

Reviewer 1 Report

In this paper the authors provided an alternative bioengineering system for SCI treatment. They constructed a 3D-printed GelMA hydrogel biomimetic spinal cord scaffold loaded with human amniotic epithelial cells-derived neural-like cells and implanted the scaffold in the lesion of a rat full transection SCI model. The 3D cell therapy system based on hAEC-neural-like cells bio-scaffolds demonstrated significant therapeutic effects in rat SCI model, indicating its potential application in clinic. There are some concerns regarding this manuscript in which addressing them may improve the article value.

Q1: Generally, the mechanical properties of the hydrogels are the main factors affecting cell survival, neuronal differentiation, and axon formation. In this paper, GelMA hydrogel was used as biomimetic spinal cord scaffold, but the effects of hydrogel with different mechanical properties on the neuronal differentiation of hAEC-neural-like cells and axonal growth were not further investigated.

Q2: SCI is usually accompanied by persistent inflammation, which is a critical mediator in secondary damage after SCI. Therefore, inflammation is also an evaluation index in the treatment of SCI. However, the evaluation of the inflammatory response after biomimetic spinal cord scaffold implantation was not performed.

Q3: In this paper, I think an additional group should be added to the group of in vivo experimental animals, and the effect of hAEC-neural-like cells implantation alone on spinal cord injury should be investigated and compared with other groups.

Q4: Some related reports in 2022 are suggested be referenced about the SCI, such as doi: 10.1186/s40779-022-00376-1; https://doi.org/10.1016/j.ajps.2021.03.003

Q5: At present, the treatment of SCI is mainly based on cell transplantation, most of which take induced pluripotent stem cells as the main transplanted cells. What is the difference between the cells mentioned in this paper and induced pluripotent stem cells, and what is the difference in the process of inducing cell differentiation?

Q6: The writing of some units in the article was not standard, such as ml and µl. It should be mL and µL.

Author Response

We are grateful to the Editor and the Reviewers for their thorough review of our manuscript (manuscript ID bioengineering-1857195) entitled “Biomimetic spinal cord scaffold loaded with human amniotic epithelial cells-derived neural-like cells repairs spinal cord injury in rats”. We are very appreciative of the thoughtful comments made by the Editors and Reviewers. We add some new data about the mechanical properties of GelMA scaffold, immunofluorescent microscopy in vivo and biomarker detection of hAEC-neural-like cells in vitro, and have more clearly explained and discussed the key points of the relative results in manuscript as required by the Editor and Reviewers. Please find enclosed our revised manuscript and our point-by-point replies to the Reviewers’ comments. All the modifications in the manuscript are marked with red font.

All of the authors have read and concur with the submission of the revised manuscript to Bioengineering. The material has not been reported previously or under consideration for publication elsewhere. We greatly appreciate your consideration of our significantly revised manuscript, and hope that you find the revisions meet your requirements for publication in your prestigious journal.

Best regards,

Luyang Yu & Yong He

Point 1: Generally, the mechanical properties of the hydrogels are the main factors affecting cell survival, neuronal differentiation, and axon formation. In this paper, GelMA hydrogel was used as biomimetic spinal cord scaffold, but the effects of hydrogel with different mechanical properties on the neuronal differentiation of hAEC-neural-like cells and axonal growth were not further investigated.

Reply: Thanks for the comment. As suggested, we added a new figure as figure S3 in supplementary materials and explained the results in lines 321-327 of the revised manuscript. Based on the results, we showed that GM-100 GelMA (substitution degree is 1M) was more suitable for hAECs attachment and growth, as evidenced by the number and morphology of hAECs seeded on the hydrogel surface. Therefore, we chose GM-100 GelMA as the raw material for the 3D printed biomimetic scaffold.

Point 2: SCI is usually accompanied by persistent inflammation, which is a critical mediator in secondary damage after SCI. Therefore, inflammation is also an evaluation index in the treatment of SCI. However, the evaluation of the inflammatory response after biomimetic spinal cord scaffold implantation was not performed.

Reply: Thanks for the comment. As suggested, we added a new figure as figure S4 in supplementary materials and elucidate the results in line 420-426 of the revised manuscript. In addition, we also discuss the potential reasons in line 491-495 in discussion. Our results indicated that no significant difference of inflammation eliminating among FT control, Scaffold and hAEC-neural-like cells scaffold rat groups, suggesting that the biomimetic spinal cord scaffold loaded with hAEC-neural-like cells had little effect on the neuroinflammation after SCI. The scaffold-based hAEC-neural-like cells mainly act as replenished cells and neurotrophic cells for repairing SCI, but may not improve the neuroinflammation around the lesion. Thus, our future work will focus on updated bioengineering strategy for the regulation of neuroinflammation and tissue micro-environment around lesions of SCI.

Point 3: In this paper, I think an additional group should be added to the group of in vivo experimental animals, and the effect of hAEC-neural-like cells implantation alone on spinal cord injury should be investigated and compared with other groups.

Reply: Thanks for the comment. We agree with the Reviewer’s rigor on the experiment design for including hAEC-neural-like cells implantation group. However, based on the harsh tissue micro-environment around lesions after spinal cord injury proved by previous studies (Bioactive Materials 6, 2523-2534 (2021); Nature 527, S193-S197 (2015); Military Medical Research 9, 16 (2022); Asian Journal of Pharmaceutical Sciences 17, 4-19 (2022)), cell grafts hardly survived without any protection from scaffolds or immunosuppressants, especially in the acute injury phase. Moreover, it is hard to guarantee to keep the cell grafts in a relatively fixed location due to their suspension state, which might hinder the consequent therapeutic effect. Therefore, we removed the hAEC-neural-like cells implantation group from our study.

Point 4: Some related reports in 2022 are suggested be referenced about the SCI, such as doi: 10.1186/s40779-022-00376-1; https://doi.org/10.1016/j.ajps.2021.03.003

Reply: Thanks for the comment. As suggested, we have added these two articles as references 10 and 29 and thereby expanded correlated statements in “introduction” in line 61-67 of the revised manuscripts. The two reviews summarize the applications of bio-scaffolds in the tissue engineering of SCI repair and elaborate the advantages of material scaffolds in loading drugs, cellular factors and stem cell grafts, which are instructive and meaningful to our study and tissue engineering field of SCI repair.

Point 5: At present, the treatment of SCI is mainly based on cell transplantation, most of which take induced pluripotent stem cells as the main transplanted cells. What is the difference between the cells mentioned in this paper and induced pluripotent stem cells, and what is the difference in the process of inducing cell differentiation?

Reply: Thanks for the comment. As mentioned in line 49-60 and 452-457 of the manuscript, there were a series of concerns about the tumorigenicity and genetic risk for iPSCs in clinical translation, even with the autologous iPSC-derived cells (N Engl J Med 377, 792-793 (2017)) and immunogenicity (Nature 474, 212-215 (2011); Nat Biotechnol. 37(10), 1137-1144 (2019)). On the other hand, as a type of perinatal cells, hAECs were proved with the guaranteed biosafety (non-tumorigenic and low immunogenicity). Besides, there is no risk of ethic problem on hAECs applications. Thus, hAECs not only has the potential to treat SCI, but also is a safe seed cell with potentials for clinical translation.

Comparing iPSCs with hAECs from the process of inducing differentiation, the iPSCs strategy follows the classic pathway of differentiation due to the pluripotency, which can produce mature functional cell types. In regard of the cell fate switch of hAECs, we prefer to define it as trans-differentiation into cells with feature of neuroepithelial lineages via mesenchymal-epithelial transition-like manner, resulting in up-regulated expression of epithelial and neural markers and reduced mesenchymal property after BGVB treatment. Therefore, the hAEC-neural-like cells could be regarded as a neuroepithelial cells comparing to the iPSC-derived neural cells that are in more mature state. Our future study will focus on generating more mature neural cells derived from hAECs by optimizing the differentiation approaches and extending the differentiation time period. We included correlated discussion in line 486-502 in the revised manuscript.

Point 6: The writing of some units in the article was not standard, such as ml and µl. It should be mL and µL.

Reply: Thanks for the comment. We apologize for our nonstandard writing of some units in the main text. As suggested, we have gone through the manuscript and made several correlated revisions in the revised manuscript.

Reviewer 2 Report

In general, I think this manuscript is well organized and interesting. To further improve a quality of this manuscript, I made a few comments.

1. In Figure 1, authors showed that hAECs expressed not only an epithelial marker (E-cadherin) but also mesenchymal markers (CD73 and CD90), suggesting that hAECs have a complex feature of epithelial and mesenchymal cells. In addition, authors also showed that hAECs displayed up-regulated expression of several neuronal markers by BGVB treatment. Did hAECs increase the expression of epithelial markers (E-cadherin) following BGVB treatment? Did hAECs decrease the expression of mesenchymal markers (CD73 and CD90) following BGVB treatment?

2. Similar to the current results, previous studies also showed that certain types of stem cells had a complex feature of epithelial and mesenchymal cells and that they acquired the feature of neuroepithelial lineages presumably via mesenchymal-epithelial transition-like manner (Cell Stem Cells, 7, 718-729, 2010; Stem Cells, 33, 1962-1974, 2015; Cell Mol life Sci, 75, 3507-3520, 2018; Stem Cell Reports, 10, 890-904, 2018). Please discuss it.

3. In Figure 6e, authors described that some hAECs (pre-labeled with red fluorescence) were positive for NF-H. However, it is likely that hAECs rarely express NF-H. Please provide a better image that one can easily understand which hAECs co-express NF-H.

4. Please provide full spell of “BGVB”.

Author Response

We are grateful to the Editor and the Reviewers for their thorough review of our manuscript (manuscript ID bioengineering-1857195) entitled “Biomimetic spinal cord scaffold loaded with human amniotic epithelial cells-derived neural-like cells repairs spinal cord injury in rats”. We are very appreciative of the thoughtful comments made by the Editors and Reviewers. We add some new data about the mechanical properties of GelMA scaffold, immunofluorescent microscopy in vivo and biomarker detection of hAEC-neural-like cells in vitro, and have more clearly explained and discussed the key points of the relative results in manuscript as required by the Editor and Reviewers. Please find enclosed our revised manuscript and our point-by-point replies to the Reviewers’ comments. All the modifications in the manuscript are marked with red font.

All of the authors have read and concur with the submission of the revised manuscript to Bioengineering. The material has not been reported previously or under consideration for publication elsewhere. We greatly appreciate your consideration of our significantly revised manuscript, and hope that you find the revisions meet your requirements for publication in your prestigious journal.

Best regards,

Luyang Yu & Yong He

Point 1: In Figure 1, authors showed that hAECs expressed not only an epithelial marker (E-cadherin) but also mesenchymal markers (CD73 and CD90), suggesting that hAECs have a complex feature of epithelial and mesenchymal cells. In addition, authors also showed that hAECs displayed up-regulated expression of several neuronal markers by BGVB treatment. Did hAECs increase the expression of epithelial markers (E-cadherin) following BGVB treatment? Did hAECs decrease the expression of mesenchymal markers (CD73 and CD90) following BGVB treatment?

Reply: Thanks for the comment. As suggested, we added a new figure as figure S2 in supplementary materials and explained the results in lines 293-296 of the revised manuscript. Briefly, we detected the epithelial and mesenchymal characteristics of hAEC-neural-like cells by flow cytometry. It was indicated that the expression of epithelial marker E-cadherin was further enhanced after BGVB treatment, suggesting that the administration of BGVB conferred hAEC-neural-like cells with a stronger epithelial property. Moreover, mesenchymal characteristic was attenuated after the inducing treatment, as evidenced by the reduced expression of CD73 and CD90. Therefore, hAEC-neural-like cells demonstrated enhanced epithelial and decreased mesenchymal properties compared with hAECs.

Point 2: Similar to the current results, previous studies also showed that certain types of stem cells had a complex feature of epithelial and mesenchymal cells and that they acquired the feature of neuroepithelial lineages presumably via mesenchymal-epithelial transition-like manner (Cell Stem Cells, 7, 718-729, 2010; Stem Cells, 33, 1962-1974, 2015; Cell Mol life Sci, 75, 3507-3520, 2018; Stem Cell Reports, 10, 890-904, 2018). Please discuss it.

Reply: Thanks for the comment. As suggested, we discussed these papers in line 479-491 of the revised manuscript, correlating to the properties changes in hAEC-neural-like cells based on the basic characteristics of hAECs. These papers enriched our study and inspired our ideas from the perspective of potential fate-switch mechanism of hAEC-neural-like cells. Due to the plasticity and neural properties as we described in the current manuscript and our previous study (Acta Pharmaco-logica Sinica 39 (8), 1305-1316 (2018)), hAECs may potentially differentiate into multiple cells lineages upon different inducing conditions. Here, our results indicated the enhanced neural and epithelial properties of hAECs after BGVB treatment, suggesting that a kind of mesenchymal-epithelial transition (MET)-like manner underwent during the switch from hAECs to hAEC-neural-like cells.

Point 3: In Figure 6e, authors described that some hAECs (pre-labeled with red fluorescence) were positive for NF-H. However, it is likely that hAECs rarely express NF-H. Please provide a better image that one can easily understand which hAECs co-express NF-H.

Reply: Thanks for the comment. As suggested, we replaced image of figure 6e with another ones (both low power and high power) to show the significant co-expression of NF-H and hAEC-neural-like cells, as indicated by white arrow heads. In the new figure, we showed the axons projected through the micro-channel filled with hAEC-neural-like cells. And the co-localization of fluorescence signals suggested the implanted hAEC-neural-like cells could be involved in the re-establishment of neural circuits and replenishment of the neural cells in the SCI lesion.

Point 4: Please provide full spell of “BGVB”.

Reply: Thanks for the comment. We apologize for our unclear statement of BGVB abbreviation in the main text. “BGVB” is an initials abbreviation (BDNF, GDNF, Vitamin C, B27 supplement) of our inducing treatment based on the neurotrophic factors and molecules combination used in this study. We also elucidated BGVB in line 94-96 and 282 of the revised manuscript. 

Reviewer 3 Report

In this study, Chen Qiu and colleagues explored the ability of a 3D printed GelMA hydrogel scaffold with micro-channels loaded with human amniotic epithelial cells-derived neural-like cells to promote functional recovery, in a complete transection rat model of spinal cord injury.

General comment

Overall, the results are promising. Still, the manuscript lacks important data necessary to assure results reproducibility (see specific comment 2). Data concerning the efficiency of the neural induction protocol used is also missing. Specifically, the expression of neural stem cell markers observed at gene expression level (real-time PCR) and at the protein level (immunofluorescence) should be supported by flow cytometry analysis of characteristic neural stem cell markers (or by quantitative analysis of immunofluorescence images), to get insight into the percentage of cells committed into a neural lineage. Conclusions are also not sufficiently supported by results. For instance, cell survival post transplantation is only supported by a single image with pre-labelled cells. The reduced number of experimental animals used to assess locomotor function and MEP response (N=3 for conditions with cells) also compromises the significance of results. Finally, in discussion, the results obtained (including those of the neural induction procotol) are poorly discussed and the novelty not emphasized.

Specific comments

1) The rationale for using a sample size of 5 (N = 5) in animal studies is not provided. For instance, did authors performed power analysis? This is relevant since a sample size of 5 is small considering the severity of the animal model used.

2) The source of a few reagents/materials (for instance gelatin, methacrylate, BDNF, GDNF) is missing. The methodology used in the following analysis is also missing: i) quantification of NGF and BDNF by ELISA; ii) pPCR analysis; iii) Flow cytometry. The pre-labelling of transplanted cells, referred in results, is also missing.

3) Why did authors used two-way ANOVA to analyze the results shown in Fig S1? The effect of a single independent variable is being assessed, namely “method”.

4) The BBB score data usually analyzed via Two-way ANOVA repeated measures. Were the BBB scores presented in Fig 5 d measured repeatedly in the same animal along time? If so, two-way ANOVA repeated measures should be performed.

5) English needs some editing/revision. See for instance line 117 “Seed for several times to avoid cells attaching on surfaces facing in one direction” in methods.

Author Response

We are grateful to the Editor and the Reviewers for their thorough review of our manuscript (manuscript ID bioengineering-1857195) entitled “Biomimetic spinal cord scaffold loaded with human amniotic epithelial cells-derived neural-like cells repairs spinal cord injury in rats”. We are very appreciative of the thoughtful comments made by the Editors and Reviewers. We add some new data about the mechanical properties of GelMA scaffold, immunofluorescent microscopy in vivo and biomarker detection of hAEC-neural-like cells in vitro, and have more clearly explained and discussed the key points of the relative results in manuscript as required by the Editor and Reviewers. Please find enclosed our revised manuscript and our point-by-point replies to the Reviewers’ comments. All the modifications in the manuscript are marked with red font.

All of the authors have read and concur with the submission of the revised manuscript to Bioengineering. The material has not been reported previously or under consideration for publication elsewhere. We greatly appreciate your consideration of our significantly revised manuscript, and hope that you find the revisions meet your requirements for publication in your prestigious journal.

Best regards,

Luyang Yu & Yong He

Point 1: The rationale for using a sample size of 5 (N = 5) in animal studies is not provided. For instance, did authors performed power analysis? This is relevant since a sample size of 5 is small considering the severity of the animal model used.

Reply: Thanks for the comment. In the preliminary experiments, we performed power analysis on sample size. The sample size of 5 achieved a power > 0.8, considering significant difference of BBB score comparing Scaffold + hAEC-neural-like cells Group with other groups (P < 0.05). Therefore, we used N = 5 in the animal studies afterwards.

Point 2: The source of a few reagents/materials (for instance gelatin, methacrylate, BDNF, GDNF) is missing. The methodology used in the following analysis is also missing: i) quantification of NGF and BDNF by ELISA; ii) pPCR analysis; iii) Flow cytometry. The pre-labelling of transplanted cells, referred in results, is also missing.

Reply: Thanks for the comment. As suggested, we added the source of materials and relative methodology in the part of 2.2 and 2.3 and line 123-154 of the revised manuscript, including 2.4. Flow Cytometry, 2.5. Enzyme-Linked Immunosorbent Assay, 2.6. Quantitative Real-Time PCR and 2.7. Pre-labelling of hAECs. Besides, we also added the primers information as table S1 in supplementary materials to provide more detailed information.

Point 3: Why did authors used two-way ANOVA to analyze the results shown in Fig S1? The effect of a single independent variable is being assessed, namely “method”.

Reply: Thanks for the comment. We apologize for our inapposite statistical method in figure S1c, d. As suggested, we revised the statistical analysis and analyzed the data again with the unpaired t-test in the new figure S1. We also did correlated revisions in the figure legend. In figure S1c and d, we compared the inducing effects among different methods and detected by the expression changing of some neural genes to confirm the optimized inducing approach, which means t-test should be used to analyze the statistical difference.

Point 4: The BBB score data usually analyzed via Two-way ANOVA repeated measures. Were the BBB scores presented in Fig 5 d measured repeatedly in the same animal along time? If so, two-way ANOVA repeated measures should be performed.

Reply: Thanks for the comment. We apologize for our unclear statement and misleading labels in figure 5d. In the analysis of BBB score, we compared the statistical difference from indicated groups in the same time point based on the mean SEM of 3 biological replicates. The asterisks we labeled in the curve of fig. 5d was from comparing Scaffold + hAEC-neural-like cells Group with FT control Group and Scaffold + hAEC-neural-like cells Group with Scaffold Group. To make the label clearer, we added the short lines with relevant colors in front of the asterisks in the new figure 5d.

Point 5: English needs some editing/revision. See for instance line 117 “Seed for several times to avoid cells attaching on surfaces facing in one direction” in methods.

Reply: Thanks for the comment. As suggested, we revised the statement in line 158-160 of the revised manuscript. Also, the manuscript was checked by our native English-speaking colleague for language editing.

Round 2

Reviewer 2 Report

I think authors significantly improved their manuscript. 

Author Response

We are very appreciative of your reply and thoughtful comments to our manuscript.

Reviewer 3 Report

The manuscript has improved but there are issues that were not adequately addressed by authors, as detailed below:

1) In the general comments, authors were asked to include data concerning the expression of characteristic neural stem cell markers, obtained by flow cytometry analysis or by quantitative analysis of immunofluorescence images. These quantitative results are necessary to support the qualitative analysis of the expression of neural stem cell markers at the protein level, obtained immunofluorescence. These results are important to provide the percentage of cells committed into a neural lineage, and, as such, the efficiency of the neural induction protocol presented (BGVB cocktail induction treatment). Authors have included in the revised version of the manuscript the expression of characteristic epithelial and mesenchymal markers (in Figure S2b and c), but not that of characteristic neural stem markers, such as nestin.

2) In the point-by-point reply to reviewers authors have provided the rationale for using a sample size of 5 (N = 5) in the animal studies, but did not include it in the revised version of the manuscript. Please include the rationale for using such a small number of animals in methods

3) Authors were told that the scientific relevance of the results could be compromised by the reduced number of experimental animals used to assess locomotor function (N=3), but this issue is not addressed in the discussion of the revised manuscript. Although statistically significant differences were obtained using such a low number of animals (N), authors should add to discussion that BBB score results were inferred from a small number of animals, and, as such, future experiments will be necessary to assess the reproducibility of these promising results.

4) Authors were asked to detail if the BBB scores presented in Fig 5 d were measured repeatedly in the same animal along time, and to analyze the BBB score via Two-way ANOVA repeated measures. Authors did not explain if the BBB scores were measured repeatedly in the same animal at several time points. Moreover, the correspondent figure legend of the revised manuscript refers now that two-way ANOVA was used to analyze the results. Since the effect of a single independent parameter “condition” was analyzed, one cannot understand why authors refer in legends the use of two-way ANOVA instead of one-way ANOVA, to detect differences between 3 conditions at the same time point. Also in Figure 5 legend, the term “difference analysis” is not correct.

5) Conclusions are still not sufficiently supported by results. Specifically, authors cannot state that the survival of transplanted cells was “decent”, based on a single image of pre-labelled cells. As such, the following sentence in line 468 must be revised “This was further supported by the observation of decent survival of hAEC-neural-like cells in the implanted scaffold, without any immunosuppressive agent’s administration”.

6) Besides these concerns, the title is very long, and the term “spinal cord” is mentioned twice. As such, I suggest altering the title from “Therapeutic effect of biomimetic spinal cord scaffold loaded with human amniotic epithelial cells-derived neural-like cells for spinal cord injury” to a shorter one, for instance “Therapeutic effect of biomimetic scaffold loaded with human amniotic epithelial cells-derived neural-like cells for spinal cord injury” 

7) English still needs editing/revision, particularly in the following sections, in which methods are presented as an experimental protocol: 

-  line 117 -  Seed for several times to avoid cells attaching on surfaces facing in one direction”. 

- Line 115-119 – “The DLP bioprinting system can be divided into three parts: optical units, material storage units, and mechanical movement units. Dissolve the GelMA into PBS buffer to make the 15% (w/v) solution and add 0.25% (w/v) initiator lithium acylphosphinate (LAP) to confer GelMA with photo-crosslinking property. Then fill the material storage unit with liquid GelMA. According to the designed structures, the section pattern would be projected to the printing platform and liquid GelMA would crosslink layer by layer. Finally, the product would be immersed into 75% ethanol and PBS buffer respectively for further experiments”.

Line 138 – “The standard procedures are followed by the nerve growth factor (NGF)and brain-derived neurotrophic factor (BDNF) ELISA Kits (R&D Systems)”.

 Line 407 – “Based on the results of the locomotion test, histological analysis of the SCI lesions  from indicated groups was performed, on the other hand”.

 Line 477 – “there are a lot of work to do in the future, especially to achieve matured neural cells efficiently”

 Figure 5 legend: “….the colored short lines front of the asterisks represented the relative groups, and the asterisks showed  the difference corresponding time points”.

Author Response

Dear Editor,

We are grateful to the Editor and the Reviewers for their further review of our manuscript (manuscript ID bioengineering-1857195) entitled “Therapeutic effect of biomimetic scaffold loaded with human amniotic epithelial cells-derived neural-like cells for spinal cord injury”. We are very appreciative of the constructive comments made by the Editor and Reviewer 3. As required, we revised improper conclusion drawing and include additional statements in discussion, re-analyzed the BBB scores in Fig. 5d using changed statistical method, modified the title, and performed English editing and revision. Please find enclosed our revised manuscript and our point-by-point replies to the Reviewers’ comments. All the modifications in the manuscript are marked up using Track Changes.

All of the authors have read and concur with the submission of the revised manuscript to Bioengineering. The material has not been reported previously or under consideration for publication elsewhere. We greatly appreciate your consideration of our significantly revised manuscript, and hope that you find the revisions meet your requirements for publication in your prestigious journal.

Best regards,

Luyang Yu & Yong He

Replies to Reviewer 3 Comments

Point 3: Authors were told that the scientific relevance of the results could be compromised by the reduced number of experimental animals used to assess locomotor function (N=3), but this issue is not addressed in the discussion of the revised manuscript. Although statistically significant differences were obtained using such a low number of animals (N), authors should add to discussion that BBB score results were inferred from a small number of animals, and, as such, future experiments will be necessary to assess the reproducibility of these promising results.

Reply: We appreciate for the comment. As suggested, we addressed the limitations in our study about the low number of animals (N=3) for the results analysis in line 506-508, and we also discussed necessary work in the future experiments to assess the reproducibility of the current results in line 511-513.

Point 4: Authors were asked to detail if the BBB scores presented in Fig 5 d were measured repeatedly in the same animal along time, and to analyze the BBB score via Two-way ANOVA repeated measures. Authors did not explain if the BBB scores were measured repeatedly in the same animal at several time points. Moreover, the correspondent figure legend of the revised manuscript refers now that two-way ANOVA was used to analyze the results. Since the effect of a single independent parameter “condition” was analyzed, one cannot understand why authors refer in legends the use of two-way ANOVA instead of one-way ANOVA, to detect differences between 3 conditions at the same time point. Also in Figure 5 legend, the term “difference analysis” is not correct.

Reply: We appreciate for the comment and apologize for our improper statement and statistical method about the BBB scores in Fig. 5d. The BBB scores were measured repeatedly in the same rat along 8 weeks post implantation to evaluate the hindlimbs locomotion, and we compared the scores of rats among different treatment groups at the same time point. Thus, as suggested, we changed the statistical method to one-way ANOVA to re-analyze the significant differences in Fig. 5d and converted the figure to group bar chart. Accordingly, the figure legend was modified and showed in line 400-405 of the revised manuscript.

Point 5: Conclusions are still not sufficiently supported by results. Specifically, authors cannot state that the survival of transplanted cells was “decent”, based on a single image of pre-labelled cells. As such, the following sentence in line 468 must be revised “This was further supported by the observation of decent survival of hAEC-neural-like cells in the implanted scaffold, without any immunosuppressive agent’s administration”.

Reply: We appreciate for the comment and apologize for our inapposite statement. As suggested, we lowered the tone and revised the sentence to “In vivo, the observation of some pre-labeled hAEC-neural-like cells in the scaffold after implantation without any immunosuppressive agent’s administration may also suggest the immunological-adaption characteristic of hAEC-neural-like cells.” in line 479-482 of the revised manuscript.

Point 6: Besides these concerns, the title is very long, and the term “spinal cord” is mentioned twice. As such, I suggest altering the title from “Therapeutic effect of biomimetic spinal cord scaffold loaded with human amniotic epithelial cells-derived neural-like cells for spinal cord injury” to a shorter one, for instance “Therapeutic effect of biomimetic scaffold loaded with human amniotic epithelial cells-derived neural-like cells for spinal cord injury”

Reply: We appreciate for the comment. As suggested, we modified the title to “Therapeutic effect of biomimetic scaffold loaded with human amniotic epithelial cells-derived neural-like cells for spinal cord injury”.
